# Rep-MedSAM: Towards Real-time and Universal Medical Image Segmentation

Muxin Wei ⬤, Shuqing Chen (✉), Silin Wu, and Dabin Xu

School of Medicine and Health,
Harbin Institute of Technology, Harbin, China
`chenshuqing@hit.edu.cn`

**Abstract.** Medical image segmentation has been a pivotal step in clinical practice, enabling more precise analysis of medical images. MedSAM, as a medical image segmentation foundation model, has significantly extended the ability of SAM to segment a broad spectrum of different modalities of medical images and achieves excellent performance comparing specialist models. However, with a heavy image encoder, MedSAM falls short of clinical usage in terms of time efficiency. Therefore, the CVPR 2024: Segment Anything In Medical Images On Laptop Challenge addresses performance and efficiency in a task, where the model infers with only CPU. To this end, we propose Rep-MedSAM, which integrates RepViT, a mobile-friendly CNN with efficient designs of lightweight ViTs, by replacing the image encoder in MedSAM. Our method is simple but effective, including knowledge distillation from pretrained MedSAM, whole-pipeline training and fine-tuning with extra datasets. We conduct all experiments on the challenge. Our method achieved an average DSC of 85.90% and an average NSD of 87.07% on validation. As for time cost, our method shows thrilling results compared to the baseline on validation. The average time for 2D and 3D cases is 0.47s and 22.47s, respectively, with an average of 2.41s for each case. Our code is available at [GitHub](GitHub).

**Keywords:** MedSAM · Rep-ViT · Medical Images.

## 1 Introduction

Accurate and efficient segmentation is an indispensable part of clinical analysis, which entails the identifying and delineating of regions of interest (ROIs) involving various of medical targets, across an extensive spectrum of modalities. In recent years, deep learning-based methods have been widely adopted to segment medical images automatically. Specifically, many important baseline methods like U-Net [23] , limited to their task-specific nature, their performance on new tasks or different data modalities, are prone to decrease. To address this lack of generality, prompt-based foundation models such as the segment anything model (SAM) [15] render few-shot or even zero-shot learning in medical images possible. Nevertheless, trained with a large amount of natural images, SAM can

hardly be applied to medical images, as natural images are significantly different from medical images. Following SAM, MedSAM [18] performs excellently in medical images with prompts. The original MedSAM is built with three main components, including an image encoder, a prompt encoder, and a mask decoder. However, with an inadequate inference speed, primarily due to the heavy image encoder, Vision Transformer [6] MedSAM falls short of clinical usage on real-time requirements. Inspired by MobileSAM [28], the proposed baseline was distilled and replaced with a lightweight image encoder TinyViT [26] from the Med-SAM image encoder ViT by imposing the image embedding outputs to be the same as MedSAM, after which we fine-tune the whole model pipeline. Despite a remarkable improvement in inference speed, the baseline still stumbles over limited computational budgets. Therefore, the CVPR 2024: SEGMENT ANY-THING IN MEDICAL IMAGES ON LAPTOP Challenge aims at efficient and well-performing semi-automatic segmentations for multimodal medical images with bounding box prompts. During the validation and test phase, the model runs in a docker environment, where only CPU and 8G of RAM are available for inference.

Recent works for downsizing ViT have shown promising inference speed in mobile devices (e.g., TinyViT, MobileViT [20]). However, carrying a large number of parameters, many ViT-based models fail to meet resource-constrained mobile devices. To this end, we propose a simple but effective method based on RepViT [25], a pure CNN architecture utilizing a lightweight structure from both CNN and ViT. RepViT has demonstrated state-of-the-art performance on various tasks versus lightweight CNNs and ViTs, showing favourable performance and inference speed. Therefore, we use RepViT as the image encoder, replacing TinyViT in the baseline.

Our main contributions of this work can be summarized as follows:

– We propose a simple but effective pipeline using a teacher-student framework to distil knowledge from a well-trained MedSAM.
– With some small architecture and preprocessing adaptations, we significantly reduce computational and time costs during distillation.
– We evaluate the effectiveness of the proposed method in context of the challenge, where we realize performance and efficiency improvement over baseline.

## 2   Method

As illustrated in Fig. 1, our framework consist of the MedSAM model (i.e., the teacher model for distillation) and our Rep-MedSAM model (i.e., the student model) to achieve efficient learning and inference.

### 2.1   Preprocessing

The challenge dataset contains over one millions image-mask pairs, covering 11 medical image modalities and more than 20 cancer types. To attain fast,

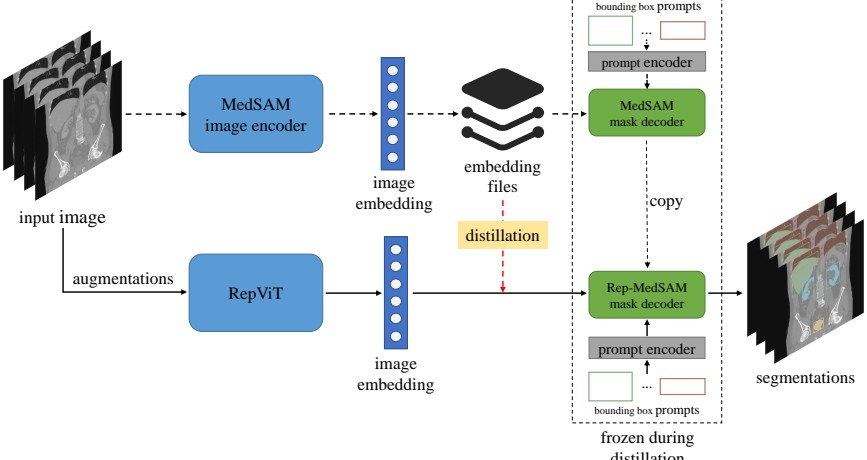

**Fig. 1.** Overview of our proposed framework.

consistent and compatible preprocessing and data loading, following [18], we make use of the following preprocessing steps:

– Dataset format conversion.
– Image intensity clipping and normalization.
– Identify non-zero slices for annotations, then crop images to non-zero regions with annotations.
– Resize all images to a uniform size of $256 \times 256 \times 3$.
– To promote distillation efficiency, we store image embeddings from the Med-SAM image encoder.

During intensity clipping, we perform specific strategies for different modalities. Notably, we normalize the Hounsfield units for CT images using typical window width and level values for specific anatomy. For grayscale images (e.g., MR, X-ray, Ultrasound, Mammography and OCT), intensity values are clipped between the 0.5th and 99.5th percentiles. As for RGB images (e.g., Dermoscopy, Endoscopy, Fundus and Pathology), if images with intensities that were within the range of $[0, 255]$, their intensities remained unaltered, otherwise normalized to the range of $[0, 255]$. In addition, for CT and MR images, we use axial slices as training inputs. Finally, to train efficiently, we resize all images to a uniform size of $256 \times 256 \times 3$.

## 2.2   Network Architecture

We inherit the mask decoder and prompt encoder from MedSAM while replacing the bulky image encoder with RepViT-M1.0. Fig. 2 shows the macro designs of

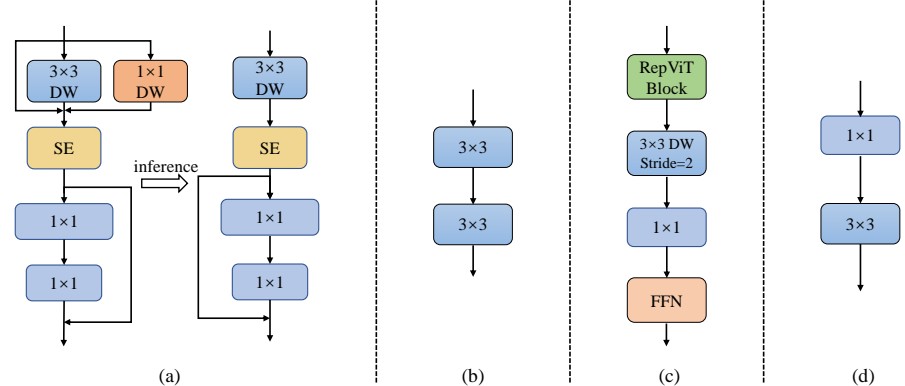

**Fig. 2. Macro designs.** (a), (b), (c) and (d) are designs of the RepViT block, stem, downsampling layer and encoding layer, respectively. Convolutional layers are simply denoted by the kernel size. The norm layer and nonlinearity are omitted.

our image encoder. Our image encoder is composed of a stem, two downsampling layers and an encoding layer. Between each layer, we adopt a ratio of 1:1:8 to insert multiple RepViT blocks. RepViT reduces computational and memory costs using the structural re-parameterization technique [5], accelerating inference speed with limited computational resources. Such a technique can reduce time cost during inference while maintaining performance. Table 6 shows the time cost with and without structural re-parameterization on the validation set.

### 2.3   Pretraining Distillation

Inspired by [26], instead of finetune-stage distillation, we pay most attention to pretraining distillation. Pretraining with distillation is insufficient and resources costly, as passing training data through the teacher model in each iteration and each epoch takes up a fair portion of computational resources. Meanwhile, a large teacher model may occupy most GPU memory, dragging the training speed down as a result of smaller batch size. To address this problem, we propose a pretraining distillation framework similar to [26,28]. However, no data augmentation is applied to training images when acquiring image embeddings from MedSAM. By contrast, we applied vertical and horizontal flipping to input images of our student model, so as to align the teacher model with better generality.

Mathematically, for an input image $x$, we store image embedding prediction $\hat{\mathbf{y}} = T(x)$ from the teacher image encoder, where $T(\cdot)$ is the teacher image encoder model. Correspondingly, we have $S(\mathcal{A}(x))$, where $S(\cdot)$ is the student image encoder and $\mathcal{A}(x)$ is the input image with data augmentation $\mathcal{A}$. During pretraining, we only need to recover $\hat{\mathbf{y}}$ from stored files, and optimize the following objective function for student model distillation:

$$\mathcal{L} = MSE(\hat{\mathbf{y}}, S(\mathcal{A}(x))), \tag{1}$$

where $MSE(\cdot)$ is the mean squared error. It is to be observed that our strategy to distil the image encoder is label-free, which enables large-scale pretraining. As depicted in Fig. 1, we copied and frozen the mask decoder from MedSAM. This is because image embeddings generated from the student image encoder are sufficiently close to the original teacher model. We conducted pretraining distillation with challenge datasets, except for the CT modality. We use only 20% of CT images, due to insufficient space, uniformly sampling from the challenge dataset to preserve data diversity.

### 2.4   Post-processing

We do not make use of any post-processing in the context of this challenge.

## 3   Experiments

### 3.1   Dataset and Evaluation Measures

We used the challenge dataset and supplementary public datasets for model development. To learn more about the supplementary datasets we used, please refer to Table 8. The evaluation metrics include two accuracy measures: Dice Similarity Coefficient (DSC) and Normalized Surface Dice (NSD)—alongside one efficiency measure—running time. These metrics collectively contribute to the ranking computation.

**Table 1.** Development environments and requirements.

| System | Ubuntu 20.04.4 LTS |
|---|---|
| CPU | Intel(R) Xeon(R) Gold 6248R@3.0GHz |
| RAM | 29GB |
| GPU (number and type) | One Nvidia Tesla V100S PCIe 32GB |
| CUDA version | 12.2 |
| Programming language | Python 3.9 |
| Deep learning framework | PyTorch (torch 2.0, torchvision 0.15.1) |
| Code | GitHub |

### 3.2   Implementation Details

**Environment Settings** The development environments and requirements are outlined in Table 1.

**Training Protocols** The training protocols for pretraining distillation, pipeline training and fine-tuning with supplementary datasets are listed in Table 2, Table 3 and Table 4. During pretraining distillation, no data augmentation is applied the to teacher model. We applied data augmentation of vertical and horizontal flipping for student model during distillation and pipeline training, combining color jitter during fine-tuning with supplementary datasets.

During pipeline training, we use all challenge datasets, while for the fine-tuning stage, we abandon CT images in the challenge dataset and train with supplementary CT datasets only. For other modalities, we trained both challenge datasets and supplementary datasets.

As mentioned in Sec. 2.3, we use MSE as the loss function during pretraining distillation. However, for the pipeline training and fine-tuning stages, we follow the loss function in baseline, using the combination of Dice loss and BCE loss for mask loss, while IoU loss for IoU between prediction and ground truth.

**Table 2.** Pretraining distillation protocols.

| | |
|---|---|
| Pre-trained Model | MedSAM [18] |
| Student Model | Rep-MedSAM |
| Data augmentation | Vertical and Horizontal Flip |
| Batch size | 16 |
| Patch size | $3 \times 256 \times 256$ |
| Total epochs | 5 |
| Optimizer | AdamW |
| Initial learning rate (lr) | 5E−4 |
| Lr decay schedule | ReduceLROnPlateau |
| Training time | 60 hours |
| Loss function | Mean squared error |
| Number of model parameters | 10.57M[1] |
| Number of flops | 468G[2] |

**Table 3.** Pipeline training protocols.

| | |
|---|---|
| Network architecture | Rep-MedSAM |
| Data augmentation | Vertical and Horizontal Flip |
| Batch size | 16 |
| Patch size | $3 \times 256 \times 256$ |
| Total epochs | 8 |
| Optimizer | AdamW |
| Initial learning rate (lr) | 3E−4 |
| Lr decay schedule | ReduceLROnPlateau |
| Training time | 128 hours |
| Loss function | Dice loss, BCE loss and IoU loss |
| Number of model parameters | 10.57M |
| Number of flops | 468G |

**Table 4.** Fine-tuning protocols.

| | |
|---|---|
| Network architecture | Rep-MedSAM |
| Data augmentation | Flip and Color Jitter |
| Batch size | 16 |
| Patch size | $3 \times 256 \times 256$ |
| Total epochs | 12 |
| Optimizer | AdamW |
| Initial learning rate (lr) | 2E−5 |
| Lr decay schedule | ReduceLROnPlateau |
| Training time | 54 hours |
| Loss function | Dice loss, BCE loss and IoU loss |
| Number of model parameters | 10.57M |
| Number of flops | 468G |

## 4    Results and Discussion

### 4.1    Quantitative Results on Validation Set

Table 5 presents the results of our Rep-MedSAM compared to the baseline on validation set at different stage. Our method manifests outstanding performance on most modalities. It is noteworthy that our method excels in Microscopy and PET images compared to the baseline. However, our method exhibits mediocre performance on PET and US images.

**Table 5.** Quantitative evaluation results on the validation set.

| Modality | Baseline | | Distillation | | Pipeline Training | | Fine-tuning | |
|---|---|---|---|---|---|---|---|---|
| | DSC(%) | NSD(%) | DSC(%) | NSD(%) | DSC(%) | NSD (%) | DSC(%) | NSD (%) |
| CT | 92.26 | 94.90 | 69.33 | 71.96 | 89.47 | 91.66 | **92.89** | **95.34** |
| MR | **89.63** | **93.37** | 78.97 | 80.78 | 81.34 | 84.25 | 87.30 | 91.06 |
| PET | 51.58 | 25.17 | 61.46 | 42.22 | **69.91** | **52.33** | 66.15 | 47.09 |
| US | **94.77** | **96.81** | 80.03 | 84.73 | 83.77 | 89.14 | 80.67 | 86.18 |
| X-Ray | 75.83 | 80.39 | 76.36 | 82.12 | 78.33 | 84.29 | **81.51** | **87.38** |
| Dermatology | 92.47 | 93.85 | 92.62 | 94.13 | 92.72 | 94.26 | **93.98** | **95.48** |
| Endoscopy | **96.04** | 98.11 | 93.54 | 96.31 | 85.77 | 89.99 | 95.67 | **98.16** |
| Fundus | 94.81 | 96.41 | 93.47 | 95.13 | 91.46 | 93.23 | **94.93** | **96.57** |
| Microscopy | 61.63 | 65.38 | 75.00 | 81.63 | 76.26 | 82.79 | **80.04** | **86.35** |
| Average | 83.22 | 82.71 | 80.08 | 81.00 | 83.23 | 84.66 | **85.90** | **87.07** |

We conducted ablation studies on efficiency between the baseline and our Rep-MedSAM with and without structural re-parameterization technique in terms of time. We built Docker images for different methods and ran under the same environment for fair comparison. The last two columns refer to the time

cost of our method without and with the structural re-parameterization technique. Compared to the baseline, both methods significantly decrease inference time by a large margin, especially in 3D cases with large volumes. Notably, when encountering multiple targets, like pathological images, it is the mask decoder which contributes the most to time cost, rather than the image encoder.

**Table 6.** Quantitative evaluation of efficiency regarding running time(s).

| Case ID | Size | Num. Objects | Baseline | w/o | w/ |
|---|---|---|---|---|---|
| 3DBox_CT_0566 | (287, 512, 512) | 6 | 436.97 | 247.97 | 194.22 |
| 3DBox_CT_0888 | (237, 512, 512) | 6 | 115.53 | 66.78 | 53.44 |
| 3DBox_CT_0860 | (246, 512, 512) | 1 | 16.60 | 9.95 | 7.96 |
| 3DBox_MR_0621 | (115, 400, 400) | 6 | 174.80 | 102.65 | 80.45 |
| 3DBox_MR_0121 | (64, 290, 320) | 6 | 119.09 | 66.01 | 54.62 |
| 3DBox_MR_0179 | (84, 512, 512) | 1 | 14.89 | 8.79 | 7.22 |
| 3DBox_PET_0001 | (264, 200, 200) | 1 | 8.94 | 5.56 | 4.86 |
| 2DBox_US_0525 | (256, 256, 3) | 1 | 0.88 | 0.47 | 0.37 |
| 2DBox_X-Ray_0053 | (320, 640, 3) | 34 | 2.19 | 1.77 | 1.59 |
| 2DBox_Dermoscopy_0003 | (3024, 4032, 3) | 1 | 1.50 | 1.15 | 0.98 |
| 2DBox_Endoscopy_0086 | (480, 560, 3) | 1 | 0.88 | 0.50 | 0.42 |
| 2DBox_Fundus_0003 | (2048, 2048, 3) | 1 | 0.89 | 0.62 | 0.48 |
| 2DBox_Microscope_0008 | (1536, 2040, 3) | 19 | 1.81 | 1.50 | 1.44 |
| 2DBox_Microscope_0016 | (1920, 2560, 3) | 241 | 14.47 | 13.79 | 13.66 |

Fig. 3 shows 4 representative segmentation results of Rep-MedSAM on unseen data, covering both large and small targets. Our method successfully segments targets in CT and MR cases. In OCT and US cases, blue arrows in the figure indicate over- and under-segmentation errors. This may be due to our model focusing too much on strong differences in intensities around ROI, neglecting ROI as a whole (e.g., the yellow box in the US case), and images losing some details after resampling.

## 4.2   Results on the Final Testing set

Table 7 shows the comparative analysis of the baseline and our proposed method across various modalities on the test set. Notably, Rep-MedSAM demonstrates significant improvements in both DSC and NSD metrics for CT scans, with an increase of approximately 17.5% and 18.5%, respectively, indicating a marked enhancement in segmentation accuracy. This trend of improved performance is consistent across most modalities, with the most substantial reduction in runtime observed in Microscopy, where our model reduces the time by over 10 seconds compared to the baseline. However, the results also highlight certain deficiencies in specific modalities, especially the X-Ray. Despite the runtime improvements in our model, the repeated computation image embeddings for 3D modalities and

the start time of the bulky docker still dominate the overall runtime, suggesting there is still room for a boost in inference and docker deployment.

**Table 7.** Quantitative evaluation results on the test set.

| Modality | Baseline | | | Rep-MedSAM | | |
|---|---|---|---|---|---|---|
| | DSC(%) | NSD(%) | Time (s) | DSC(%) | NSD (%) | Time (s) |
| CT | 55.75 | 58.48 | 38.78 | **73.21** | **76.95** | **20.65** |
| MR | 64.80 | 62.75 | 18.57 | **71.24** | **66.44** | **10.48** |
| PET | 76.94 | 66.98 | 14.9 | **80.09** | **70.59** | **9.57** |
| US | 85.24 | 89.73 | 8.96 | **89.25** | **93.48** | **5.93** |
| X-Ray | **85.51** | **94.40** | 9.95 | 78.45 | 89.25 | **5.72** |
| Endoscopy | **94.41** | **96.95** | 7.56 | 93.87 | 96.60 | **5.21** |
| Fundus | 87.47 | 89.58 | 8.77 | **84.63** | **86.87** | **5.33** |
| Microscopy | 84.36 | 86.15 | 16.34 | **88.12** | **90.02** | **5.71** |
| OCT | 73.31 | 80.20 | 8.39 | **83.11** | **89.66** | **5.4** |
| Average | 78.64 | 80.58 | 14.69 | **82.44** | **84.43** | **8.22** |

### 4.3    Limitation and Future Work

While our model outperforms the baseline in segmentation results, there is still room for further improvement. Regarding inference speed, the image encoder is the most time-consuming component when inferencing with fewer objects, whereas the mask decoder becomes the bottleneck when processing a larger number of objects. Additionally, segmenting each slice in 3D images independently leads to higher time costs and the potential loss of spatial information, which may impact segmentation accuracy. We will refer to the updated research progress to improve the quality and speed of 3D image segmentations in our future work.

## 5    Conclusion

In this paper, we proposed a framework based on knowledge distillation to leverage large pre-trained MedSAM for more efficient semi-automatic segmentations based on bounding boxes. Our method demonstrates excellent results in both efficiency and performance. Our proposed framework can serve as a robust tool for medical image segmentation in clinical practice.

**Acknowledgements** We thank all the data owners for making the medical images publicly available and CodaLab [27] for hosting the challenge platform.

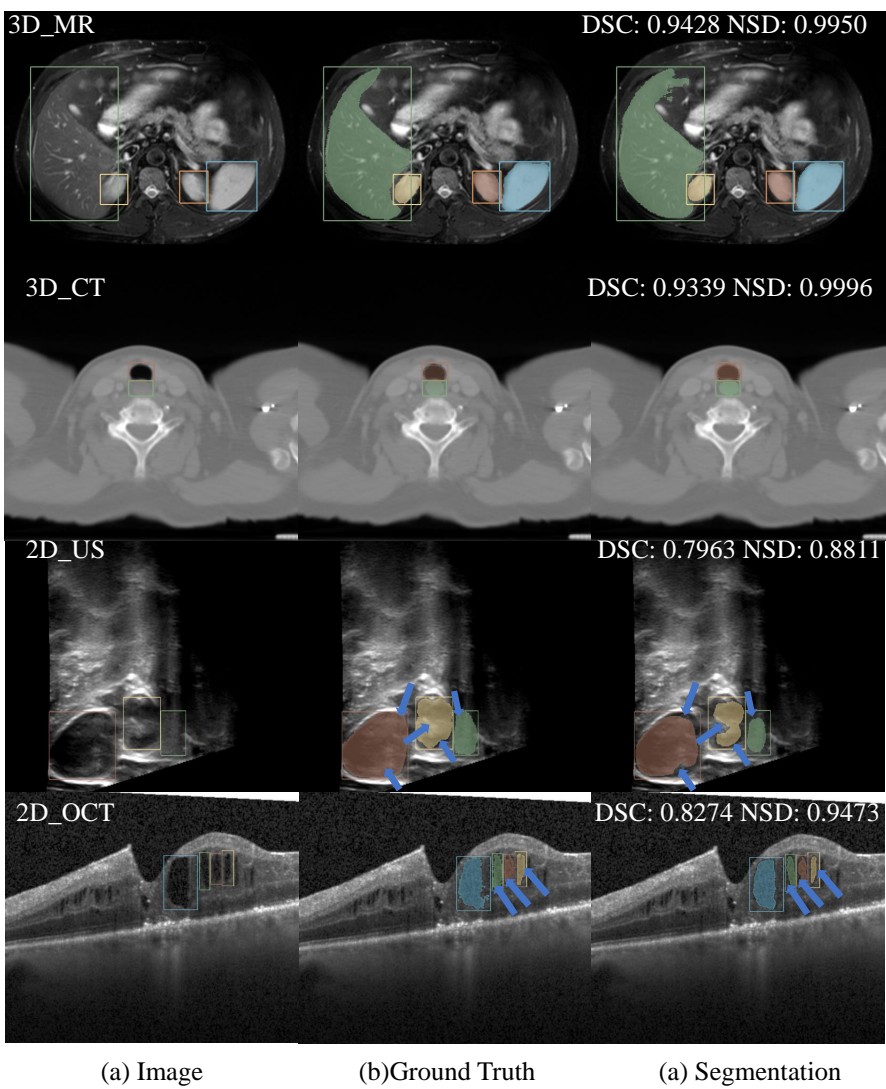



(a) Image          (b)Ground Truth          (a) Segmentation



**Fig. 3.** Qualitative results of our Rep-MedSAM on two easy cases (MR and CT) and two hard cases (US and OCT).

**Table 8.** External public datasets we used in fine-tuning.

| Modality | Dataset Name |
|---|---|
| CT | CT-Org [22], ULS23 [11], FLARE22 [19], HaN-Seg [21] |
| Dermoscopy | None |
| Endoscopy | BKAI-IGH NeoPolyp [2,16,7] |
| Fundus | E-ophtha [4] |
| Mammography | None |
| Microscopy | PanNuke [8][9] |
| MR | ARC [10], ACDC [3], CDEMRIS [14] |
| PET | None |
| Ultrasound | CAMUS [17], MicroSegNet [13], CT2US [24] |
| X-Ray | Panoramic X-ray [1], Hip joint X-ray [12], |

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

**Table 9.** Checklist Table. Please fill out this checklist table in the answer column.

| Requirements | Answer |
| --- | --- |
| A meaningful title | Yes |
| The number of authors ($\leq 6$) | 4 |
| Author affiliations and ORCID | Yes |
| Corresponding author email is presented | Yes |
| Validation scores are presented in the abstract | Yes |
| Introduction includes at least three parts: background, related work, and motivation | Yes |
| A pipeline/network figure is provided | 1 |
| Pre-processing | 2 |
| Strategies to data augmentation | 6 |
| Strategies to improve model inference | 3 |
| Post-processing | 5 |
| Environment setting table is provided | 1 |
| Training protocol table is provided | 2, 3, 4 |
| Ablation study | 8 |
| Efficiency evaluation results are provided | 6 |
| Visualized segmentation example is provided | 3 |
| Limitation and future work are presented | Yes |
| Reference format is consistent. | Yes |
| Main text $>=$ 8 pages (not include references and appendix) | Yes |