# OpenReview forum: "Rep-MedSAM: Towards Real-time and Universal Medical Image Segmentation"
_thecvf.com/CVPR/2024/Workshop/MedSAMonLaptop — CVPR24 MedSAMonLaptop_

### Official Review · Reviewer_MiUH · 2024-06-15
**Well-structured article with some issues**

**Rating:** 7
**Confidence:** 4

**Review:**

Pros:
- The paper is well-structured and provides a comprehensive overview of the development and evaluation of Rep-MedSAM. The methodology and results are promising, showcasing significant improvements in efficiency and performance.
- It's interesting that you only use the external dataset for the CT modality in the training pipeline. It would be nice if you could comment more on why you do this.

Cons:
- In the main contribution (in the introduction), "distill" is misspelled.
- In the data preprocessing section, there are 11 modalities in the training set, it seems like you have missed the microscopy type.
- In Figure 2, some blocks are just noted as "1x1" and "3x3." You should specify what those blocks are.
- In the distillation phase, the teacher model was not applied with augmentation, but the student model was. How can you align these two outputs with MSE loss while their orientations do not match?
- The scores in the abstract do not match the scores in the Table 5.
- The scores of hard cases in qualitative results are still high (0.88 and 0.94 NSD scores). You should pick more challenging cases.
- Some sections are missing: qualitative results, efficiency results, limitations, and future work.

---

### Official Review · Reviewer_qorE · 2024-06-15
**lack of descriptions on necessary content**

**Rating:** 6
**Confidence:** 5

**Review:**

1. Please add a paragraph to describe the strategies for improving inference efficiency. Even failed attempts are also worth mentioning.

2. Sec 2.4: Instead of only using one sentence as a paragraph, please merge it with other paragraphs.

3. Fig. 1-2: Please add more descriptions of the network in the caption.

4. Table 6: Explain w/o and w/ in the caption.

5. Please add a comprehensive description of the algorithm's advantages and weaknesses in discussion section.

6. Please add a paragraph on future work.

7. References: please delete the doi links

---

### Official Review · Reviewer_6gmY · 2024-06-16
**Well-structured but needs clarification on several points**

**Rating:** 6
**Confidence:** 5

**Review:**

Strengths:
* This paper is clear and thorough in presenting the development of Rep-MedSAM.
* The results are compelling, indicating that the methodology introduced by the paper can significantly improve performance relative to the baseline.

Weaknesses:
* The paper does not sufficiently explain why Rep-ViT is favored over TinyViT when both are smaller ViT variants with similar sizes (Rep-MedSAM params: 10.57 M, LiteMedSAM: 9.66 M).
* The rationale for why excluding 80% of the CT dataset could contribute to training set diversity during pretraining distillation is not well-explained.
* The term “pipeline training” is not explained within the context of this paper. This stage appears to take a significant proportion of the overall training time compared to other stages; however, the focus remains on the pretraining distillation. Additionally, what distinguishes pipeline training from finetuning? Were any modules frozen during finetuning?
* While the paper highlights improved inference speed with the reparameterization technique of Rep-ViT in the result section, it does not address whether this technique affects the segmentation performance in terms of DSC and NSD scores on the challenge's tasks. Including an ablation study on this aspect could strengthen the paper.
* In Figure 3’s caption, it should be “qualitative results” instead of “quantitative results.” Additionally, including the baseline’s visualized segmentation results in the comparison would provide a clearer benchmark for evaluating improvements.
* Typo: In the first paragraph of Introduction, it should be “after which we fine-tune the whole pipeline.”

---

### Official Review · Reviewer_no7Z · 2024-06-21
**Great results, but lacks novelty, analysis, and extensive ablations.**

**Rating:** 7
**Confidence:** 5

**Review:**

The quality of the paper was reasonable. The key idea (significance) was to use a RepViT backbone instead of a TinyViT.

The authors mentioned most details that are necessary for implementation. Some of the missing details include mentioning which particular variant of RepViT was used and defining what "whole pipeline training means".

The writing of the document was clear and formal. The only suggestion I would have is to replace the bullet points on page 3 with an numerically ordered list.

The paper's training process and architecture was not novel, as it is similar to MobileSAM and traditional knowledge distillation. Some novel ideas include storing the teacher embeddings, training the student's prediction on augmented images to match teacher's prediction on the unaugmented images (during distillation and whole-pipeline training),  and cropping the training images to non-zero regions with annotations.

The ablation study only compared computational speed of structural reparamerization technique, which I beleive was important. Further ablations should include the changes in performance (e.g. comparing the improvement in performance from the novel ideas).

The analysis of the sample results was very limited.

In terms of training improvements, I would look into whether distilling the student and teacher on different views is actually beneficial, as there are works that indicate the contrary: "Knowledge distillation: A good teacher is patient and consistent" (https://arxiv.org/abs/2106.05237).


Measuring the carbon footprint of the training (CO2)

---

### Decision · Program_Chairs · 2024-10-01

Accept